# XREF: Entity Linking for Chinese News Comments with Supplementary Article Reference

**Xinyu Hua**                                              HUA.X@HUSKY.NEU.EDU
*Northeastern University*
*Boston, MA 02115*

**Lei Li**                                               LILEILAB@BYTEDANCE.COM
*ByteDance AI Lab,*
*Beijing, China*

**Lifeng Hua**                                       ISSAC.HLF@ALIBABA-INC.COM
*Alibaba Group,*
*Hangzhou, China*

**Lu Wang**                                                LUWANG@CCS.NEU.EDU
*Northeastern University*
*Boston, MA 02115*

## Abstract

Automatic identification of mentioned entities in social media posts facilitates quick digestion of trending topics and popular opinions. Nonetheless, this remains a challenging task due to limited context and diverse name variations. In this paper, we study the problem of entity linking for Chinese news comments given mentions' spans. We hypothesize that comments often refer to entities in the corresponding news article, as well as topics involving the entities. We therefore propose a novel model, XREF, that leverages attention mechanisms to (1) pinpoint relevant context within comments, and (2) detect supporting entities from the news article. To improve training, we make two contributions: (a) we propose a supervised attention loss in addition to the standard cross entropy, and (b) we develop a weakly supervised training scheme to utilize the large-scale unlabeled corpus. Two new datasets in entertainment and product domains are collected and annotated for experiments. Our proposed method outperforms previous methods on both datasets.

## 1. Introduction

Social media, including online discussion forums and commenting systems, provide convenient platforms for the public to voice opinions and discuss trending events [O'Connor et al., 2010, Lau et al., 2012]. Entity linking (EL), which aims to identify the knowledge base entry (or the lack thereof) for a given mention's span, has become an indispensable tool for consuming the enormous amount of social media posts. Concretely, automatically recognizing entities can quickly inform who and what are popular [O'Connor et al., 2010, Zhao et al., 2014, Dredze et al., 2016], promote semantic understanding of social media content, and facilitate downstream tasks, such as relation extraction, opinion mining, questions answering, and personalized recommendation [Messenger and Whittle, 2011, Galli et al., 2015]. Although EL has been extensively investigated in newswire [Kazama and Torisawa, 2007, Ratinov et al., 2011], web pages [Demartini et al., 2012], and broadcast news [Benton and

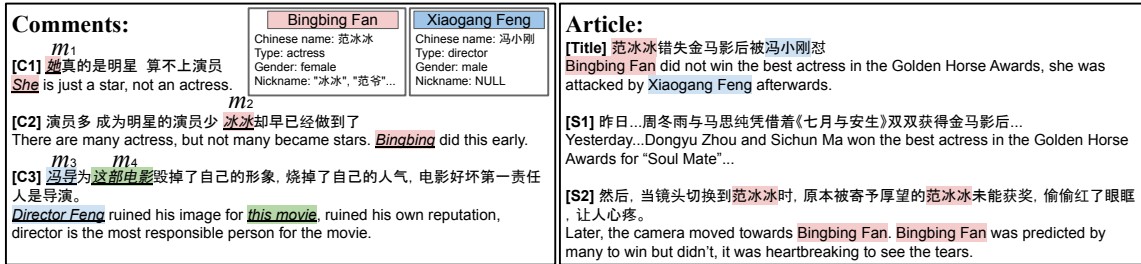

Figure 1: Sample user comments with entity mentions underlined. $m_1$ (pronoun) and $m_2$ (nickname) are linked to "Bingbing Fan". Mention $m_3$ (unknown nickname) refers to "Xiaogang Feng". Their associated entities can be inferred from the article, but not from the comment alone.

Dredze, 2015], its study in the social media domain was only started more recently [Guo et al., 2013a, Yang and Chang, 2015, Moon et al., 2018], mostly focusing on English.

In this paper, *we study the task of entity linking for user comments in online Chinese news portals.* To the best of our knowledge, we are the first to investigate EL problem for the genre of news comments at a large scale. Besides issues present in the conventional EL work [Ji et al., 2010], social media text poses additional challenges: *the lack of context* and *increased name variations due to its informal style.* State-of-the-art EL methods [Francis-Landau et al., 2016, Gupta et al., 2017] heavily rely on modeling the text surrounding the mentions, as the abundant context from longer documents greatly helps identify entity related content. However, context is often scant for user comments. For instance, as shown in Figure 1, the entity mention $m_2$ may indicate "Bingbing Fan" or "Bingbing Li", both being prominent actresses. By looking up the entities covered in the article, which contains unambiguous mention of the former entity, an EL system will be more confident to link $m_2$ to it. Moreover, the informal style and evolving vocabulary on social media lead to enormous name variations based on aliases, morphing, and misspelling. For instance, in one of our newly annotated datasets, the maximum number of distinct mentions of an entity is 121.

In this work, we propose XREF, *a novel entity linking model for Chinese news comments by exploiting context information of entity mentions as well as identifying relevant entities in reference articles.* XREF, with its overview displayed in Figure 2, has three key properties. First, we enrich the mention representation with two sources of information through attentions. **Comment attention** pinpoints topics involving the target entity from comment context. For instance, words "star" and "actress" in comment $C1$ in Figure 1 provide useful information about entity types. **Article entity attention** detects target entities from the articles if they are discussed. Furthermore, we investigate a new objective function to drive the learning of article entity attention. Finally, we also exploit data augmentation with distant supervision [Mintz et al., 2009] to leverage large amounts of unlabeled comments and articles for model training.

Since there was no publicly-available annotated dataset, as part of this study, we collect and label two new datasets of Chinese news comments from the domain of entertainment and product, which are crawled from a popular Chinese news portal `toutiao.com`.[1] Experimental results show that our best performing model obtains significantly better accuracy

---

1. Datasets and code can be found at `http://xinyuhua.github.io/Resources/akbc20/`.

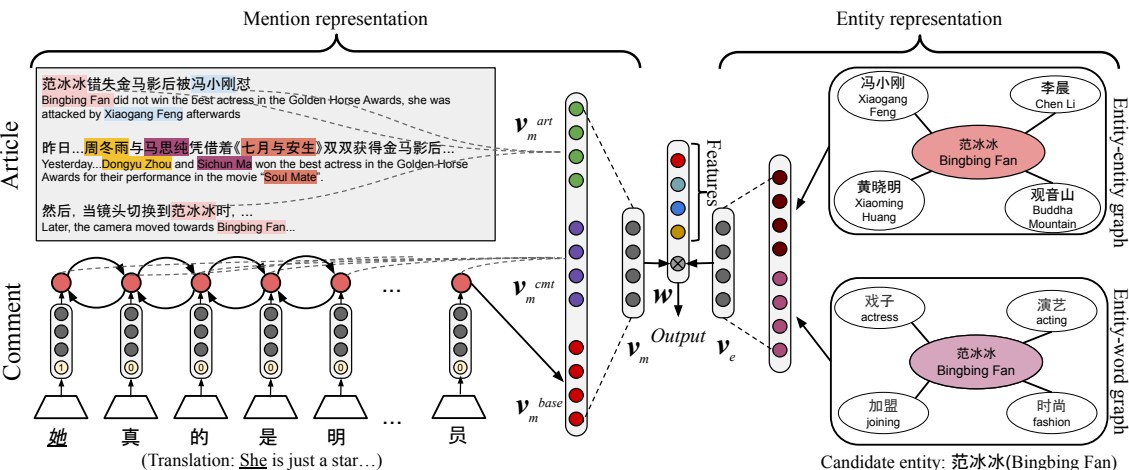

Figure 2: Overview of XREF model. It learns to represent mentions (left) and entities (right). Context-aware mention representation encodes the information about comment ($v_m^{base}$ and $v_m^{cmt}$) and article ($v_m^{art}$) via attention mechanisms. Entity representation is built on entity-entity and entity-word co-occurrence graph embeddings. The dot product of the mention representation and entity representation can be concatenated with a feature vector to produce the final output after a layer of linear transformation.

and Mean Reciprocal Rank scores than the state-of-the-art [Le and Titov, 2018] and other competitive comparisons. For example, our model improves the accuracy by at least three points over the state-of-the-art model in both domains with NIL mentions considered (67.2 vs 58.6 on entertainment comments, and 77.3 vs 68.6 on product comments).

## 2. Related Work

Entity linking (EL), as a fundamental task for information extraction, has been extensively studied for long documents, such as news articles or web pages [Ji et al., 2010, Shen et al., 2015]. State-of-the-art EL systems rely on extensive resources for learning to represent entities with diverse information, including entity descriptions given by Wikipedia or knowledge bases [Kazama and Torisawa, 2007, Cucerzan, 2007], entity types and relations with other entities [Bunescu and Paşca, 2006, Hoffart et al., 2011b, Kataria et al., 2011], and the surrounding context [Ratinov et al., 2011, Sun et al., 2015]. Neural network-based models are designed to learn a similarity measure between a given entity mention and previously acquired entity representation [Francis-Landau et al., 2016, Gupta et al., 2017, Le and Titov, 2018]. However, very limited context is provided in social media posts. In this work, we propose to leverage attention mechanisms to identify salient content from both comments and the corresponding articles to enrich the entity mention representation.

Our work is inline with the emerging entity linking research for social media content [Liu et al., 2013, Guo et al., 2013a,b, Fang and Chang, 2014, Hua et al., 2015, Yang and Chang, 2015]. To overcome the lack of context, existing models mostly resort to including extra information, e.g., considering historical messages by the same authors or socially-connected authors [Guo et al., 2013b, Shen et al., 2013, Yang et al., 2016], or leveraging posts of

|                            | Entertainment | Product |
| -------------------------- | ------------- | ------- |
| # News Articles            | 10,845        | 8,275   |
| Avg # Sents per Article    | 18.5          | 13.5    |
| Avg # Chars per Sentence   | 40.4          | 50.7    |
| # Comments                 | 967,763       | 410,790 |
| Avg # Chars per Comment    | 21.3          | 23.2    |
| # Annotated Comments       | 30,630        | 5,189   |
| # Annotated Mentions       | 46,942        | 7,497   |
| # Annotated Unique Entities| 1,846         | 470     |

Table 1: Statistics of crawled datasets from entertainment and product domains.

similar content [Huang et al., 2014]. However, users in news commenting systems might be anonymous, and few additional posts would be available for newly published articles. We therefore study a more practical setup, without using any of the aforementioned information as input.

## 3. Data Collection and Annotation

We collect user comments along with corresponding news articles from `toutiao.com`, a popular Chinese online news portal. A sample article snippet with comments is displayed in Figure 1. Two popular domains are selected for annotation: entertainment (ENT) and product (PROD). Articles and comments in ENT focus on movies, TV shows, and celebrities, whereas most topics in PROD are automobiles and electronic products. The statistics of the crawled dataset after filtering are in Table 1. As illustrated, there are only an average of 20 characters in a comment, highlighting the lack of context.

**Annotation Procedure.** We randomly sample 995 articles from ENT and 783 articles from PROD, and annotate the corresponding user comments. Articles and comments that are not in the samples are used for model pre-training via data augmentation (§ 4.6).

Annotators are presented with both comments and corresponding articles during the annotation process. They first identify mention spans, where named, nominal, and pronominal mentions of entities are labeled. Each mention is then linked to an entity in a knowledge base, or labeled as NIL if no entry is found. Though not in our knowledge base, the word "小编" (editor) is included as an entity due to its popularity. We also allow one mention to be linked to multiple entities, e.g. plural pronoun "他们" (they/them). Comments without any mention are discarded. 13 professional annotators, who are native Chinese speakers with extensive NLP annotation experience, are hired, each annotating a different subset. An additional human annotator conducts the final check.

**Statistics.** Final statistics for the datasets are displayed in Table 1. On average, there are 4.4 distinct mentions per entity, with a maximum number of 121 for domain ENT. For PROD, the average mention number is 2.9 with a maximum number of 48. Sample mentions are shown in Table 2.

We categorize the samples into the following types, based on the entities mentioned by: (1) **canonical names** as defined in knowledge base; (2) **nicknames** as the popular aliases included in the knowledge base for each entity; (3) **pronominal** mentions indicating one

| Entity (uniq. mentions) | Sample Mentions |
|---|---|
| 范冰冰 (121)
"Bingbing Fan" | "戏子(actress)", "冰姐 (sister Bing)", "范 (Fan)", "国际女神 (international goddess)" |
| 那英 (111)
"Ying Na" | "戏子 (actress)", "满族后裔 (descendant of Manchu people)", "自个 (herself)", "演员 (actress)" |
| 别克英朗 (48)
"Buick Excelle" | "手动精英 (stick shift elite)", "这款车 (this car)", "我的车子 (my car)", "2016款英朗 (2016 Excelle)" |
| 马自达3昂克赛拉 (47)
"Mazda3 Axela" | "两厢 (hatchback)", "自动舒适型 (automatic and comfortable)", "昂克塞拉 (Axela)", "昂克赛拉1.5自动舒适车 (Axela 1.5T automatic)" |

Table 2: Entities with the most unique mentions from entertainment and product domains.

| | Canon. | Nick. | Pron. | Others | Plural | NIL |
|---|---|---|---|---|---|---|
| **Entertainment** | 29.8% | 4.0% | 12.9% | 21.9% | 2.9% | 28.4% |
| **Product** | 33.9% | 0.6% | 2.7% | 41.1% | 0.2% | 21.6% |

Table 3: Mention type distribution.

entity, such as "他 (he/him)" or "这个 (this)"; (4) **plural pronominal** mentions that are linked to multiple entities; (5) **others**, all other types of mentions that can be linked to the KB, including aliases not in the knowledge base or misspellings; and (6) **NIL**, mentions that cannot be linked to any entity in the KB. The mention type distributions are in Table 3, and it is observed that pronominal and nickname mentions are more common in Ent where celebrities are frequently discussed. The type of *others* is more significant in Prod due to the prevalent usage of irregular name variations for products.

**Knowledge Base.** Baidu Baike[2], a large-scale Chinese online encyclopedia, is used to construct the knowledge base (KB). A snapshot of Baike containing 68,067 unique entities was collected on May 10th, 2017. Four attributes are leveraged for feature engineering: (1) *gender*, (2) *nicknames* as a list of common aliases for an entity, (3) *entity type*, and (4) *entity relation*.

## 4. The Proposed Approach

Our model takes as input a phrasal mention $m$ in a comment $C$, which is posted under an article. Given a knowledge base, we aim to predict the KB entity that $m$ refers to, or to label it as NIL if no such entity exists. Concretely, a list of candidate entities will be first selected as $\mathcal{E}_m = \{e\}$ based on string matching and knowledge graph expansion (see § 4.1). Then a linking probability will be computed over each candidate given $m$.

### 4.1 Candidate Construction

Our candidate construction algorithm consists of two steps. For each mention, we consider all entities that appear in the same comment and corresponding article by matching their canonical names. This forms the initial candidate list. In the second step, a new entity

---

2. https://baike.baidu.com

is selected if it has a relation with any entity in the initial list according to our KB. The initial list, the expanded entities, and NIL comprise the final candidate set.

Following this procedure, 96% gold-standard entities are retrieved in the candidate sets for ENT, and 62% are covered for PROD. To improve the coverage for the PROD domain, we collect unambiguous aliases (no other entity with the same alias) that are not pronominal mentions from training data for each entity, and use these as additional entity nicknames for candidate construction. The coverage is increased to 93%.

## 4.2 Entity Representation

Prior work for entity representation learning usually relies on entity-word co-occurrence statistics derived from the entities' English Wikipedia pages [Francis-Landau et al., 2016, Gupta et al., 2017, Ganea and Hofmann, 2017, Eshel et al., 2017]. Unfortunately, Wikipedia has low coverage of entities in our newly collected Chinese datasets. We thus consider two sources of information, both acquired from news headlines. First, a graph-based node2vec [Grover and Leskovec, 2016] embedding $\boldsymbol{u}^{nod}$ is induced from an *entity-entity* co-occurrence matrix extracted from 65 million news titles after applying canonical name matching [Zwicklbauer et al., 2016, Yamada et al., 2016]. $\boldsymbol{u}^{nod}$ is expected to capture entity relations. Second, a Singular Vector Decomposition (SVD)-based representation $\boldsymbol{u}^{wrd}$ is obtained from an *entity-word* co-occurrence matrix constructed from the same set of news titles. We concatenate them as $\boldsymbol{u}$ and apply a one-layer feedforward neural network over it to form the entity representation $\boldsymbol{v}_e = \tanh(\boldsymbol{W}_e\boldsymbol{u} + \boldsymbol{b}_e)$, where $\boldsymbol{W}_e \in \mathbb{R}^{300 \times 600}$ and $\boldsymbol{b}_e \in \mathbb{R}^{300 \times 1}$ are trainable parameters.

## 4.3 Mention Representation

We train character embeddings from the 327 million user comments with word2vec [Mikolov et al., 2013]. A bidirectional Long Short-Term Memory (biLSTM) network is then applied over comment character embeddings $\boldsymbol{x}_i^c$, with hidden state $\boldsymbol{h}_i = [\overrightarrow{\boldsymbol{h}_i}; \overleftarrow{\boldsymbol{h}_i}]$ for each time step $i$. We append a one-bit mask $q_i$ to the character embeddings to indicate the mention span. If a character is within the mention span, $q_i$ is 1; otherwise, it is 0. $\boldsymbol{h}_i$ is calculated recurrently as $\boldsymbol{h}_i = g(\boldsymbol{h}_{i-1}, [\boldsymbol{x}_i^c; q_i])$, where $g$ is the 200-dimensional biLSTM network. The last hidden state $\boldsymbol{h}_T$ is taken as the base form of mention representation $\boldsymbol{v}_m^{base}$.

**Comment Attention.** Preliminary studies show that $\boldsymbol{v}_m^{base}$ focuses on the local context, and does not capture long-distance information well. Hence we propose to learn an importance distribution over all comment characters through a bilinear attention [Luong et al., 2015] with query $\tilde{\boldsymbol{m}}$, the average character embeddings of the mention:

$$\tilde{\boldsymbol{m}} = \frac{1}{m_e - m_s} \sum_{i=m_s}^{m_e} \boldsymbol{x}_i^c \tag{1}$$

$$\alpha_i^{cmt} = \frac{\exp(\boldsymbol{h}_i^T \boldsymbol{W}_c \tilde{\boldsymbol{m}})}{\sum_{i'=1}^{T} \exp(\boldsymbol{h}_{i'}^T \boldsymbol{W}_c \tilde{\boldsymbol{m}})} \tag{2}$$

$$\boldsymbol{v}_m^{cmt} = \sum_{i=1}^{T} \alpha_i^{cmt} \boldsymbol{h}_i \tag{3}$$

where $m_s$, $m_e$ are start and end offsets of the mention span. $\boldsymbol{x}_i^c$ is the character embedding of the $i$-th character in the comment, $\boldsymbol{W}_c \in \mathbb{R}^{200 \times 300}$ is the trainable bilinear matrix.

**Article Entity Attention.** Intuitively, users tend to comment on entities covered in the news. We thus design an article entity attention to identify target entities if they appear in the article, or indicate non-existence otherwise. Concretely, articles are segmented into words by Jieba[3], an open source Chinese word segmentation tool. Each word is matched with canonical entity names in the knowledge base, and the article is represented as a set of unambiguous entities, $\mathcal{E}_a$. Each entity is represented as $\boldsymbol{u} = [\mathbf{u}^{nod}; \mathbf{u}^{wrd}]$. We also add one *absent padding entity* (denoted as **ABS**), a 300-dimension zero vector, into the set to indicate that the entity is not in the article. The article entity representation $\boldsymbol{v}_m^{art}$ is calculated as:

$$\beta_j^{art} = \text{softmax}(\boldsymbol{u}_j^T \boldsymbol{W}_a \tilde{\boldsymbol{m}}) \tag{4}$$

$$\boldsymbol{v}_m^{art} = \sum_{j=1}^{|\mathcal{E}_a|} \beta_j^{art} \boldsymbol{u}_j \tag{5}$$

where $\boldsymbol{u}_j$ is the entity representation for $j$-th entity in $\mathcal{E}_a$. $\boldsymbol{W}_a \in \mathbb{R}^{600 \times 300}$ is the bilinear matrix parameter.

### 4.4 Learning Objective

XREF learns to align the mention representation $\boldsymbol{v}_m$ and the candidate entity representation $\boldsymbol{v}_e$ after transforming them into a common semantic space. Specifically, the base form $\boldsymbol{v}_m^{base}$, comment attended $\boldsymbol{v}_m^{cmt}$, and article attended $\boldsymbol{v}_m^{art}$ are concatenated as the input to a feedforward neural network to form $\boldsymbol{v}_m$ as $\boldsymbol{v}_m = \tanh(\boldsymbol{W}_m[\boldsymbol{v}_m^{base}; \boldsymbol{v}_m^{cmt}; \boldsymbol{v}_m^{art}] + \boldsymbol{b}_m)$. Given a mention $m$ represented as $\boldsymbol{v}_m$, the probability for $m$ being linked to an entity $e$ (represented as $\boldsymbol{v}_e$) is computed by applying the softmax function over the dot product between their representations, over all candidates in $\mathcal{E}_m$: $P(e|m) = \text{softmax}_{e \in \mathcal{E}_m}(\boldsymbol{v}_e \cdot \boldsymbol{v}_m)$ The entity with the highest positive likelihood is selected as prediction. Previous work [Yang et al., 2016] has found that surface features can further improve representation learning-based EL models. We thus append **features** (§ 4.5) to the dot product via $P(e|m) = \text{softmax}_{e \in \mathcal{E}_m}\left(\mathbf{w} \cdot [\boldsymbol{v}_e \cdot \boldsymbol{v}_m; \boldsymbol{\Phi}(m)]\right)$, where $\boldsymbol{\Phi}(m)$ is the feature vector and $\mathbf{w}$ are learnable weights.

During training time, we use the same candidate construction algorithm in § 4.1 to collect negative samples, where all candidates except the gold-standard are treated as negative. The cross-entropy loss on training set is defined as:

$$\mathcal{L}_{EL}(\theta) = -\sum_n \sum_k y_{n,k}^* \log(P(e_k|m_n)) \tag{6}$$

where $P(e_k|m_n)$ is the predicted probability for the $k$-th entity candidate for $n$-th mention in training set. $y_{n,k}^*$ represents the gold-standard, it has a value of 1.0 for positive samples, and 0.0 for negative ones.

---

3. https://github.com/fxsjy/jieba

**Supervised Attention Loss.** Notice that the article entity attention naturally learns an alignment between the mention and entity representation $\boldsymbol{u}$. To help learn high quality alignment, we design a new learning objective to provide direct supervision to the article entity attention. To the best of our knowledge, we are the first to design supervised attention mechanism to guide entity linking. Concretely, during training, if an entity in $\mathcal{E}_a$ matches the gold-standard, we assign a relevance value of 1.0 to it; otherwise, the score is 0.0. If none from $\mathcal{E}_a$ matches, the absent padding entity is labeled as relevant. We thus design the following objective for article attention learning:

$$\mathcal{L}_{Att}(\theta) = -\sum_n \sum_j \beta_{n,j}^* \log(\hat{\beta}_{n,j}) \tag{7}$$

$\beta_{n,j}^*$ is the true relevance value for $j$-th article entity, and $\hat{\beta}_{n,j}$ is the attention calculated as in Eq. 4, both are extended with mention index $n$ (i.e. the $n$-th mention in the training set). The **final learning objective** becomes $\mathcal{L}(\theta) = \mathcal{L}_{EL}(\theta) + \lambda \cdot \mathcal{L}_{Att}(\theta)$. $\lambda$ is set to 0.1 in all experiments below.

### 4.5 Features

We optionally append 20 features to the output layer, as detailed in Table 4, where the last 11 features are adopted from [Zheng et al., 2010].

### 4.6 Weakly Supervised Pre-training

We leverage the unlabeled samples for data augmentation. Concretely, mentions and entities are automatically labeled if an entity's canonical name or nickname can be matched in a comment unambiguously (i.e., no other entity with the same name). In total, this procedure automatically labeled $502,858$ comments for the ENT domain, which is split into $453,080$ for training and $49,778$ for validation. For the PROD domain, we create $175,951$ comments, among which $158,336$ are for training and $17,615$ are for validation. Each dataset is used to pre-train XREF, which is then trained on the annotated data.

## 5. Experimental Setup

Each dataset is split into training, validation, and test sets based on articles, with statistics displayed in Table 5. Articles in test sets are published later than those in training and validation sets. For this study, we focus on the task of entity linking, therefore gold-standard mention spans are assumed to have been provided. A mention detection component will be developed in future work.

**Hyperparameters.** For all experiments, Adam optimizer [Kingma and Ba, 2015] is used with an initial learning rate of 0.0001. We adopt gradient clipping with a maximum norm of 5. Model batch size is set to 128.

**Baselines.** We design five baselines: (1) MATCHCANON matches the mention with canonical names in KB, and outputs an entity if a match is found, otherwise predicts NIL; (2) MATCHCANONANDNICK further matches nicknames if MATCHCANON returns NIL, (3) FREQUENCYINART predicts the most frequent entity in the article; (4) FIRSTINART pre-

| Feature | Description |
|---|---|
| CanonMatch | Whether the mention text exact-match the canonical KB name |
| NicknMatch | Whether the mention text exact-match the canonical KB name |
| CharJaccard | The char-level Jaccard score btw. the mention and entity's canonical name |
| PinyJaccard | The Jaccard similarity between the Pinyin of the mention and entity's canonical name |
| GendMatch | Whether the gender of pronominal mention matches that in KB |
| EntArtFreq | The frequency of candidate entity in article, considering both exact canonical name searching and nickname searching |
| CommentDist | The distance between the canonical name of the candidate entity and the mention in the comment, if the canonical name is not present set to 100 |
| PriorProb | probability $P(e\|m)$ with MLE |
| Special | Whether the mention is a domain-specific entity, such as "小编" (editor) |
| EditDist | The edit distance between mention and entity on character level |
| StartWithMent | Whether any of the entity's canonical name or nickname starts with the mention string |
| EndWithMent | Whether any of the entity's canonical name or nickname ends with the mention string |
| StartInMent | Whether any of the entity's canonical name or nickname is a prefix of the mention string |
| EndInMent | Whether any of the entity's canonical name or nickname is an affix of the mention string |
| EqualWordCnt | The maximum number of same words between mention and entity's canonical name and nicknames |
| MissWordCnt | The minimum number of different words between mention and entity's canonical name and nicknames |
| ContxtSim | TF-IDF similarity between entity's Baike article and comment |
| ContxtSimRank | Inverted rank of ContxtSim across all candidates |
| AllInSrc | Whether all words in candidate entity's canonical name exist in comment |
| MatchedNE | The number of matched named entities between entity's Baike page and comment |

Table 4: Features used in our model and comparisons.

| | Train | | Valid | | Test | |
|---|---|---|---|---|---|---|
| | article | comment | article | comment | article | comment |
| **Entertainment** | 734 | 23,046 | 98 | 3,153 | 149 | 4,415 |
| **Product** | 587 | 3,943 | 78 | 473 | 118 | 773 |

Table 5: Experimental setup statistics.

dicts the first entity in the article; (5) PriorProb predicts the most likely entity based on $P(e|m)$, estimated from entity-mention co-occurrence in the training set.

**Comparisons.** We further compare against the following models: (1) Vector Space Model (VSM) computes TF-IDF cosine similarity between mention context and entity KB pages [4], with the most similar candidate as prediction. (2) Logistic Regression (LogReg) trained with features described in the next paragraph. (3) ListNet is a learning-to-rank approach that outperforms all methods in the EL track of TAC-KBP2009 [Zheng et al., 2010]. (4) CEMEL expands mention representation with similar posts and then applies VSM [Guo et al., 2013b]. We retrieve all comments containing the mention string from the training set as similar posts. (5) Ment-norm [Le and Titov, 2018] is the state-of-the-art EL model on

---

4. We include all content from entity's Baidu Baike page.

| | Entertainment | | | | Product | | | |
|---|---|---|---|---|---|---|---|---|
| | *with NIL mentions* | | *w/o NIL mentions* | | *with NIL mentions* | | *w/o NIL mentions* | |
| | **Acc** | **MRR** | **Acc** | **MRR** | **Acc** | **MRR** | **Acc** | **MRR** |
| **Baselines** | | | | | | | | |
| MATCHCANON | 42.70 | - | 37.91 | - | 54.35 | - | 44.00 | - |
| MATCHCANONANDNICK | 44.66 | - | 40.22 | - | 54.97 | - | 44.78 | - |
| FREQUENCYINART | 32.66 | - | 35.67 | - | 16.34 | - | 20.44 | - |
| FIRSTINART | 29.17 | - | 31.86 | - | 17.41 | - | 21.78 | - |
| PRIORPROB | 54.22 | 57.16 | 51.65 | 54.85 | 77.71 | 78.44 | 76.89 | 77.80 |
| **Learning-based Models** | | | | | | | | |
| VSM | 26.04 | 34.70 | 28.44 | 37.90 | 33.57 | 42.72 | 42.00 | 53.45 |
| LOGREG | 57.69 | 62.76 | 63.00 | 68.54 | 61.37 | 63.11 | 76.78 | 78.96 |
| LISTNET | 55.44 | 59.22 | 60.55 | 64.67 | 61.55 | 63.18 | 77.00 | 79.05 |
| CEMEL | 33.01 | 39.48 | 36.05 | 43.11 | 44.23 | 50.31 | 55.33 | 62.94 |
| MENT-NORM | 58.60 | 63.69 | 60.53 | 65.50 | 68.56 | 71.32 | 79.22 | 81.81 |
| **XREF** (Ours) | **67.22**$^*$ | **73.92**$^*$ | **69.84**$^*$ | **75.46**$^*$ | **77.26** | **81.52** | **81.56** | **83.94** |

Table 6: Entity linking results on singular mentions with and without NIL (non-existence in KB) considered. The best performing learning-based models are highlighted in **bold** per column. No MRR result is reported for baselines where only one entity is returned. Our models that are statistically significantly better than all the baselines and comparisons are marked with $*$ ($p < 0.0001$, approximation randomization test [Noreen, 1989]).

AIDA-CoNLL [Hoffart et al., 2011a], which consists of English news articles. It leverages latent relations among mentions to find global optimal linking results. Important parameters of MENT-NORM, such as the number of latent relations, are tuned on our development set. The same entity and character embeddings as in our model are utilized.

## 6. Results and Analysis

**Main Results.** We report evaluation results based on accuracy and Mean Reciprocal Rank (MRR) [Voorhees et al., 1999], which considers the positions of gold-standard entities ranked by each system. Table 6 displays evaluation results for entity mentions excluding plural pronominal mentions. We experiment with two setups based on whether NIL is considered for training and prediction.

Overall, our model achieves significantly better results than all other comparisons on the ENT domain for both setups ($p < 0.0001$, approximation randomization test). For PROD domain, our model also obtains the best accuracy and MRR when NIL is not included. When NIL is considered, while the strong baseline based on prior probability $p(e|m)$ achieves marginally better accuracy, our model yields higher MRR. This is because the ENT domain has much more pronominal mentions (12.9%) than the PROD domain (2.7%). On ENT, our models perform especially well at resolving pronominal cases; on PROD, the prior baseline memorizes the names better, yet our model still obtains the best MRR when NIL is considered.

**Results on Plural Pronominal Mentions.** Though rarely studied in prior work [Ji et al., 2016], it is common to observe pronominal mentions linked to multiple entities in social media. Here we report results on plural pronominal mentions only in Table 7. We assume the true number of entities is given as $K$, which varies among samples; top $K$ candidates

|  | **Acc@**$K$ | **MRR** | **NDCG** |
|---|---|---|---|
| Listnet | 1.97 | 31.70 | 42.59 |
| Ment-norm | 4.72 | 33.98 | 34.74 |
| LogReg | 12.99 | 55.07 | 60.69 |
| **XREF** w/ Art Attn | **30.31**$^*$ | **63.76**$^*$ | **68.61**$^*$ |

Table 7: Results on plural pronominal mentions for Ent domain. $K$ indicates the number of entities in the gold-standard. Significant better results than all comparisons are marked with $*$ ($p < 0.0001$, approximation randomization test).

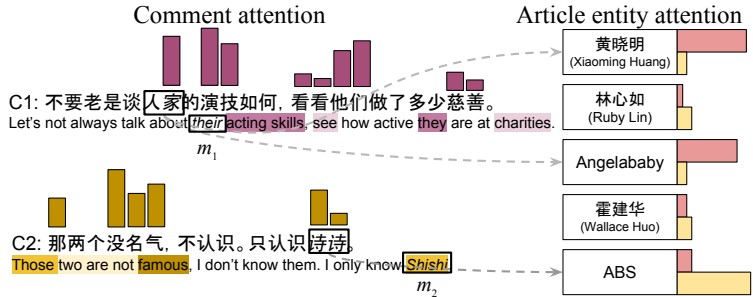

Figure 3: An illustration of comment and article attentions. Attention over comment characters is depicted by color shading. Note that the article attention successfully selects both entities for $m_1$ ("their") (pink histograms). ABS: absent padding entity, indicating entity not in article.

output by each model are compared against the gold-standards. In addition to accuracy@$K$ and MRR, Normalized Discounted Cumulative Gain (NDCG) [Järvelin and Kekäläinen, 2002] that considers multiple target predictions is reported. As can be seen, XREF with article entity attention significantly outperforms other comparisons. This is likely because plural nominal often refers to the entities in the article, suggesting the effectiveness or article attentions in these samples. Further experiments show that the full model with additional comment attention and features actually yields marginally lower scores.

We further show sample comment attention and articles entity attention output by our model in Figure 3. For the plural pronominal mention ("their") in comment $C1$, the article entity attention correctly identifies both "Xiaoming Huang" and "Angelababy" from the news. Comment attention also pinpoints phrases related to the entities, e.g., "acting skills" and "charities". For mention $m_2$, the article entity attention also correctly indicates entity's non-existence in the article by giving a high weight to the absent padding entity.

**Error Analysis.** We break down the errors made by each model based on different *mention types*, as illustrated in Figure 4. Our model XREF produces much less errors in pronominal mentions and other name variations than the comparisons. However, the name matching-based baseline achieves better performance on canonical mentions, indicating a future direction for designing better representation learning over names.

**Effect of Data Augmentation and Ablation Study.** We examine the effect of data augmentation by evaluating models that are trained with manually labeled data only. As can be seen in Table 8, for both domains, there are significant accuracy drops. Moreover,

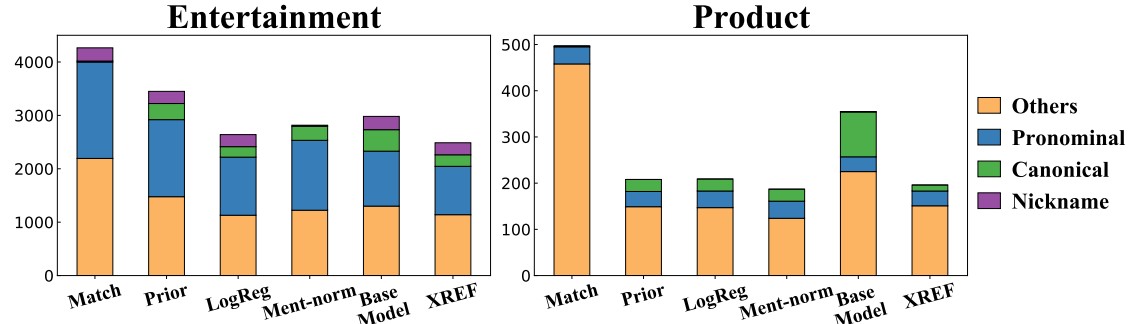

Figure 4: Error breakdown based on mention type. Our model makes less errors on pronominal mentions and name variations (Others) not captured by KB. "Base Model" represents XREF without attentions and features.

|  | Entertainment | Product |
| --- | --- | --- |
| **XREF** | 57.93 | 65.33 |
| w/o Comment Attn | 51.94 | 62.22 |
| w/o Comment + Article Attn | 44.90 | 55.33 |

Table 8: Accuracy by our models without data augmentation (NIL not considered).

accuracy drops further when attentions or features are removed. This again demonstrates the effectiveness of comment attention and article entity attention proposed by this work.

## 7. Conclusion

We present a novel entity linking model, XREF, for Chinese online news comments. Attention mechanisms are proposed to identify salient information from comments and corresponding article to facilitate entity resolution. Model pre-training based on data augmentation is conducted to improve performance. Two large-scale datasets are annotated for experiments. Results show that our model significantly outperforms competitive comparisons, including previous state-of-the-art. For future work, additional languages, including low-resource ones, will be investigated.

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
