# OpenReview forum: "XREF: Entity Linking for Chinese News Comments with Supplementary Article Reference"
_AKBC.ws/2020/Conference — AKBC 2020_

### Official Review · AnonReviewer1 · 2020-03-28
**decent approach, new datasets**

**Rating:** 7
**Confidence:** 4

**Review:**

This paper focuses on the problem of entity linking in Chinese social media. Compared to entity linking in documents, entity linking in social media poses additional problems as social media post have limited context and the language is informal. The paper proposes XREF, which overcomes these problems by utilizing additional context from comments and associated articles, and by using data augmentation. The paper is overall well written.

XREF uses an attention mechanisms to pinpoint relevant context within comments, and detect supporting entities from the news article. A weakly supervised training scheme is utilized to employ unlabelled corpus. The authors also propose two new low-resource datasets. Experimental results demonstrate effectiveness of XREF over other baselines.

The paper would have been stronger if results on at least one more language were reported.  Discussion/comparison with the following relevant prior work will be useful.

1. Entity Linking on Chinese Microblogs via Deep Neural Network, Weixin Zeng, Jiuyang Tang, Xiang Zhao
2. Chinese Social Media Entity Linking Based on Effective Context with Topic Semantics, Chengfang Ma, Ying Sha, Jianlong Tan, Li Guo, Huailiang Peng

---

> ### Author Response · Authors · 2020-04-16
> **Response to reviewer 1**
>
> Thank you for your valuable comments! Here are our response to your comments.
>
> > The paper would have been stronger if results on at least one more language were reported.
>
> We agree. Although this work only studies Chinese social media, most of our model components are language independent (with the exception of certain features that rely on Chinese phonetic similarity), and therefore we expect this framework to adapt well to other languages.
>
> > Discussion/comparison with the following relevant prior work will be useful.
>
> Thanks for the reference. We will add discussions in our revision.

---

### Official Review · AnonReviewer3 · 2020-03-30
**Entity linking across news and comments in Chinese with promising results**

**Rating:** 7
**Confidence:** 3

**Review:**

The paper describes a method to perform entity linking across news and news comments, in Chinese, using attention mechanisms to pinpoint relevant context within comments and detect supporting entities in the article body. The authors use a weakly supervised training scheme to work with a large scale corpus.

The method is well described, the model has promising results compared to the state of the art, and the Chinese-language entity linking corpus is a welcome addition. Because of these reasons, the paper is a good candidate for the conference.

The only suggestion I have for the camera-ready version is a discussion about the generalizability of this methodology. Is this method dependent on the article-comment structure? Would it work with other datasets, e.g. a Wikipedia page and editor discussions?

Finally, I have a question about the usage of attention. Would it make sense to use other comments in addition to the article body itself for the detection of supporting entities? It seems like this could help in the case when conversations happen between commenters.

---

> ### Author Response · Authors · 2020-04-16
> **Response to reviewer 3**
>
> Thank you for the valuable comments! Here are our response to your questions.
>
> > The only suggestion I have for the camera-ready version is a discussion about the generalizability of this methodology. Is this method dependent on the article-comment structure? Would it work with other datasets, e.g. a Wikipedia page and editor discussions?
>
> Because our framework consists of multiple modules (i.e., article and comment attention, features, candidate entity selection), some of which are domain-independent. We believe it could work in many other domains with moderate change to the architecture. The news comment domain is arguably more difficult than Wikipedia and editor discussions due to the lack of context and informal naming variations. Therefore our model design is crafted to exploit the additional information from the articles.
>
> > Would it make sense to use other comments in addition to the article body itself for the detection of supporting entities? It seems like this could help in the case when conversations happen between commenters.
>
> This would be an interesting next step. We do observe that comments under the same article offer complementary information to each other. And we could leverage the confident prediction to consolidate the less certain ones.

---

### Official Review · AnonReviewer4 · 2020-03-31
**An effective model for entity-linking that leverages a related news article for context**

**Rating:** 7
**Confidence:** 4

**Review:**

Summary: This work presents a novel neural model, XREF, for entity linking in Chinese online news comments. Two new datasets of news articles and comments in entertainment and product domains are collected and annotated for evaluation on this task. The unique problem setup-up facilitates XREF to use the corresponding news article in the following ways:
(a) construct a candidate entity set with high coverage since comments mostly discuss entities in the article;
(b) use a novel attention mechanism over the news article;
(c) Guide these article attention values using a supervised loss;
Furthermore, XREF leverages unannotated articles and comments using match-based weak supervision. The model achieves improvements over existing SOTA entity linking models and strong baselines for the proposed tasks, especially for plural pronominal mentions.

Pros: Authors identify a novel way to tackle the lack of context for entity-linking in social media posts: use the corresponding news article connected to the posts. The work presents an effective model to use a related/linked article when it's available. There is potential to combine XREF with other sources of context like user history for broader applications.

Cons:
- Comments without entities are left out in the constructed datasets. This could make the task of mention detection harder as negative samples are missing.
- The paper lacks ablations for weak supervison and the different attention mechanisms proposed (comment, article)

---

> ### Author Response · Authors · 2020-04-16
> **Response to reviewer 4**
>
> Thank you for the valuable comments! Here are our response to the two cons:
>
> > Comments without entities are left out in the constructed datasets. This could make the task of mention detection harder as negative samples are missing.
>
> The comments without annotated entities are indeed useful, and we do leverage a good amount of them to pre-train the character embeddings (sec 4.3). We also want to clarify that the focus of this paper is Entity Linking, therefore the mention spans are given as input. In the future, we plan to expand our framework to be able to handle mention detection as well.
>
> > The paper lacks ablations for weak supervison and the different attention mechanisms proposed (comment, article)
>
> Thanks for pointing this out! We did experiment with ablated models on different attention mechanism and non data augmentation models. These are omitted in the current manuscript due to space limitations, we will include them in the revision.

---

### Decision · Program_Chairs · 2020-04-30

**Decision:**

Accept

**Comment:**

All reviewers are fairly positive about the paper that deals with entity linking in Chinese online news comments. The key strengths of the paper are: using additional context from associated articles by using data augmentation, novel attention mechanism over news articles, guidance to article attention values. The paper is well written and has good results over state of the art. Reviewers pointed out suggestions for further work like trying another language, doing ablation study, testing the generalizability, etc. While all of these are good ideas to make the work more comprehensive and thorough, still the paper stands on its own merits, and should be a good addition to the conference.